# Bioaccumulation of Microcystin-LR and Induced Physio-Biochemical Changes in Rice (*Oryza sativa* L.) at Vegetative Stage under Hydroponic Culture Conditions

**DOI:** 10.3390/toxins16020082

**Published:** 2024-02-04

**Authors:** Jinlin Jiang, Yue Shi, Feng Tian, Tao Long, Xuzhi Li, Rongrong Ying

**Affiliations:** 1State Environmental Protection Key Laboratory of Soil Environmental Management and Pollution Control, Nanjing Institute of Environmental Sciences, Ministry of Ecology and Environment of China, Nanjing 210042, China; jl_jiang@zju.edu.cn (J.J.);; 2College of Defense Engineering, Army Engineering University, Nanjing 210007, China

**Keywords:** *Oryza sativa* L., microcystin-LR, bioaccumulation, physio-biochemical index, superoxide anion, nitric oxide synthase

## Abstract

Irrigation with water containing a variety of microcystins (MCs) may pose a potential threat to the normal growth of agricultural plants. To investigate the phytotoxicity of MC-LR at environmental concentrations on rice (*Oryza sativa* L.), the characteristics of uptake and accumulation in plant tissues, as well as a series of key physio-biochemical process changes in leaves of rice seedlings, were measured at concentrations of 0.10, 1.0, 10.0, and 50.0 μg·L^−1^ in hydroponic nutrient solutions for 7, 15, 20, and 34 days. Results showed that MC-LR could be detected in rice leaves and roots in exposure groups; however, a significant accumulation trend of MC-LR in plants (BCF > 1) was only found in the 0.10 μg·L^−1^ group. The time-course study revealed a biphasic response of O_2_^•−^ levels in rice leaves to the exposure of MC-LR, which could be attributed to the combined effects of the antioxidant system and detoxification reaction in rice. Exposure to 1.0–50.0 μg·L^−1^ MC-LR resulted in significant depletion of GSH and MDA contents in rice leaves at later exposure times (15–34 days). Low MC-LR concentrations promoted nitric oxide synthase (NOS) activity, whereas high concentrations inhibited NOS activity during the later exposure times. The reduced sucrose synthase (SS) activities in rice exposed to MC-LR for 34 days indicated a decrease in the carbon accumulation ability of plants, and therefore may be directly related to the inhibition of plant growth under MC exposure. These findings indicate that the normal physiological status would be disrupted in terrestrial plants, even under exposure to low concentrations of MC-LR.

## 1. Introduction

In recent years, there has been growing evidence that harmful cyanobacterial blooms are increasing in frequency, magnitude, and duration globally [1,2,3]. Cyanobacterial blooms have influenced freshwater ecosystems worldwide during the last few decades. Common taxa in the bloom-forming cyanobacteria species (BFCS) include those from the genera of *Microcystis*, *Anabaena*, *Nodularia*, and *Cylindrospermopsis*, etc. [3,4,5]. As the most common BFCS, *Microcystis* has attracted great attention because of its toxic secondary metabolites of microcystins (MCs). MCs are potent hepatotoxins in higher organisms, with over 90 different variants [1,6], the most toxic and prevalent of which are microcystin-LR (MC-LR), MC-RR, and MC-YR [7,8]. During bloom outbreaks, the dissolved MC concentrations in water normally range between 0.1 and 10 μg·L^−1^, while the ecological risks of low MC concentrations in natural environments remain unclear, and the risks that are associated with MCs in the food chain have not been fully elucidated [1,2,9]. Moreover, MCs can enter farmland via irrigation water [10,11] and induce phytotoxicity in terrestrial plants [12,13,14,15,16]. Previous research has already shown that MCs can accumulate in the plant body via spray irrigation [17]. Up to 8.10 ± 4.67 μg·kg^−1^ DW (dry weight) of MCs were detected in *Capsicum annuum* and *Solanum lycopersicum* irrigated with water containing cyanobacterial blooms [18]. Chen et al. [13] found that MC-LR concentrations in rice seed samples collected in the Lake Taihu area ranged from 0.04 to 3.19 μg·kg^−1^. The uptake and accumulation of cyanobacterial toxins in plants will inevitably trigger a series of physiological and biochemical reactions that can inhibit plant growth, causing the reduction of the quality and productivity of terrestrial agricultural plants [19].

MCs specifically inhibit protein phosphatases 1 and 2A in plant and animal cells, which leads to a series of downstream effects [20,21,22]. The toxicity mechanism mediated by oxidative stress in plants has also been well characterized in recent years [1,9,15,23]. Exposure to 0.50–10.0 μg·L^−1^ of MCs in the form of cyanobacteria crude extract or pure toxin could induce the responses of the antioxidant systems of plants, as evidenced by significant changes in the activities of superoxide dismutase (SOD), peroxidase (POD), catalase (CAT), glutathione peroxidase (GPX), and glutathione S-transferase (GST) [24]. Jiang et al. [1] also demonstrated that glutathione (GSH) was involved in the detoxification of MC-LR in *Vallisneria natans*, and oxidative damage induced in plants was demonstrated by a significant increase in the malondialdehyde (MDA) content at 1.0 μg·L^−1^ of dissolved MC-LR. Nitric oxide (NO) interacts in different ways with reactive oxygen species (ROS) and may play a role as an antioxidant under certain stress conditions [25,26,27]. NO is mainly synthesized and released by nitric oxide synthase (NOS) and nitrate reductase, among which inducible nitric oxide synthase (iNOS) plays a dominant transcriptional role and synthesizes most of the cellular NO [28]. So far, the relationship between MC-LR exposure and NO generation has not been reported in plants.

It has been demonstrated that low concentrations of MC-LR can promote the increase of rice biomass (including plant height, root length, and fresh weight), while, at higher concentrations, MC-LR plays an inhibitory role on vegetative growth, leaving the plants small, with yellow leaves [10]. The mechanism of how MCs influence crop growth has not been well elucidated [11]. Sucrose synthase (SS) is a glycosyl transferase enzyme that plays a key role in sugar metabolism of plants. Plants with reduced SS activity have been shown to have reduced growth, reduced starch, cellulose, or callose synthesis, reduced tolerance to anaerobic-stress conditions, and altered shoot apical meristem function and leaf morphology [29]. Based on the above background, the current work aims to delve further into the characteristics of uptake and accumulation in leaf and root tissues, as well as a series of key physio-biochemical process changes in leaves of rice seedlings under long-term exposure to MC-LR.

## 2. Results and Discussion

### 2.1. Levels of MC-LR in Rice Leaves and Roots

Using the enzyme-linked immunosorbent assay (ELISA), accumulated MC-LR was measured in rice leaves. Table 1 shows the MC-LR detected in leaf and root tissues of all exposed plants groups. For the 7-day experimental group and the 50.0 μg·L^−1^ MC-LR group for 15 days, we did not collect enough leaves to determine the MC-LR content. By day 20, the level of MC-LR in leaf tissues increased with increasing environmental concentration from 0.1 to 10 μg·L^−1^ MC-LR. At a given exposure concentration of MC-LR, there was an initial increase in the MC-LR detected in rice leaves, followed by a decrease in general. The highest accumulation of MC-LR in leaves tended to occur around the 20-day exposure period. The MC-LR content in rice roots was positively correlated with the concentration of MC-LR in the nutrient solution after exposure for 7 days (*r* = 0.917). We did not see a good linear relationship between the exposure concentration and the amount of MC-LR in rice roots at subsequent time periods. The amount of accumulated MC-LR in rice roots decreased with time (15–34 d) in the 10.0 μg·L^−1^ MC-LR group. In the groups of 15- and 20-day treatments, the root MC-LR levels in rice exposed to 50.0 μg·L^−1^ MC-LR were significantly reduced compared with those in groups with lower concentrations of MC-LR (*p* < 0.05).

This study calculated the relative accumulation of MC-LR in the rice roots and leaves. According to Table 1, the ratio of MC-LR in plant tissues (leaves or roots) to water in the lowest concentration treatment groups (0.1 μg·L^−1^) were both greater than 1, suggesting that the rice would uptake and accumulate the MC-LR in leaves and roots. However, the ratios in the treatment groups with higher concentrations (10.0, 50.0 mg·kg^−1^) were much lower than 1, indicating that the living rice did not actively uptake a significant amount of MC-LR for self-defense.

In ecosystems, MCs show bioaccumulation in zooplankton, molluscs and crustaceans, fish, and other common invertebrates and vertebrates. Animal cells have special membrane transport proteins, *OsOATPM*, that play a part in toxin accumulation [30], but a transporter protein has not been confirmed for MC-LR in terrestrial plants. Recent research has explored how *OsOATPM* aids in the exocytosis of MC-LR in plants [19,31]. Several reports on the bioaccumulation of MCs in aquatic plants (*Lemna gibba* and *Vallisneria natans*) suggest that terrestrial plants, including food crops, can also accumulate MCs through irrigation water [1,32]. By irrigating with eutrophic water containing cyanobacterial blooms and dissolved toxins, terrestrial plants may become exposed to cyanobacterial toxins. Chen et al. [12] exposed rice to 24 μg·L^−1^ MC-LR for 10 days and did not detect any MC in rice leaves, while the same concentration resulted in higher bioaccumulation in exposed rapeseed plant leaves. Maejima et al. [33] found that broccoli plants exposed to 0.01–10 μg·mL^−1^ MC-RR for 7 days did not accumulate MC-RR if they were exposed to concentrations lower than 0.1 μg·mL^−1^. The bioaccumulation in broccoli exposed to 1, 5, and 10 μg·mL^−1^ of MC-RR was 14.5, 88.9, and 145 ng·mL^−1^, respectively. Moreover, MC-LR was discovered for the first time to bioaccumulate in rice grain in the Lake Taihu region. MC-LR was detected in 21 of the total 44 sited rice grain samples, with a range in the concentration from 0.04 to 3.19 μg·kg^−1^ (dry weight) [13]. According to our study, the bioaccumulation of MC-LR in rice leaves and roots does not follow a linear relationship with environmental MC-LR concentrations or time of exposure. Our results show that, after the same period, the MC-LR in plants tissues increased with environmental concentration, but there was a sudden decrease in MC-LR in the high concentration exposure group (50.0 μg·L^−1^). The BCF values of toxin in leaves and roots indicated that the bioaccumulation of MC-LR in the plant tissues and roots only happened in the lowest concentration treatment groups (0.10 μg·L^−1^). The MC-LR bioaccumulation, exposure concentration, exposure time, and plant type are all interrelated. When the plant is exposed to higher concentrations of MC-LR, its defense mechanisms are activated. The bioaccumulated MC-LR in rice might be partially bio-transformed through a glutathione-related pathway, which is also the important detoxification pathway in aquatic plants [4,23]. The plant can transfer the conjugate of GSH-MCs into vacuoles for storage and further processing. Therefore, the cell structure damages in plant root cells caused by high concentrations of MC-LR exposure may favor the MC-LR accumulation in plant tissue to a certain extent [19,34]. In addition, the active transport mechanism of toxins in plants may also be affected by a high concentration of MC-LR stress.

### 2.2. Effects of MC-LR Exposure on O_2_^•−^ Levels in Rice Leaves

ROS are often associated with environmental stress-related plant toxicity. Figure 1 depicts the levels of O_2_^•−^ in rice leaves following MC-LR exposure. During early exposure times, the rice plants were still in the seedling stage and provided inadequate sample sizes. For this reason, we began measuring the levels of O_2_^•−^ after 10 days of MC-LR exposure, which was also more reflective of changes in enzymatic activities. As shown in Figure 1, after 10 days of MC-LR exposure, all groups had significantly lower levels of O_2_^•−^ than the control group (*p* < 0.05, *p* < 0.01). The levels of O_2_^•−^ in the 0.10, 1.00, 10.0, and 50.0 μg·L^−1^ MC-LR groups, on the other hand, were significantly higher than those in the control group (*p* < 0.05, *p* < 0.01) after 20 days of exposure, with increases of 60.7%, 27.7%, 114%, and 90.1%, respectively. By day 34, overall O_2_^•−^ levels appeared to be decreasing, with O_2_^•−^ levels in the 1.0 and 10.0 μg·L^−1^ MC-LR groups significantly lower when compared with the control group (*p* < 0.05).

ROS are produced in vivo under normal physiological conditions, reacting chemically with biological macromolecules, including unsaturated fatty acids, proteins, and DNA [1,2,34]. These reactions result in damage to macromolecules, which affects their normal physiological functions and induces oxidative stress. Under normal physiological conditions, there is a dynamic equilibrium between the ROS production and the antioxidant defense system of the organism. MCs have been shown in studies to induce the excessive production of ROS in aquatic plant tissues directly, and the ROS levels in a plant are closely related to environmental stress-induced toxicity [1,23]. The results of our study demonstrate that MC-LR indeed directly induces excessive O_2_^•−^ production in rice leaves. As shown in Figure 1, after 10 days of MC-LR exposure, the concentration of O_2_^•−^ decreased with increasing MC-LR concentrations, reaching a minimum when exposed to 50.0 μg·L^−1^ MC-LR. This is likely due to the early responses of the antioxidant system to neutralize the excessive amounts of ROS. After 20 days of exposure, plants exposed to 0.10, 10.0, and 50.0 μg·L^−1^ MC-LR had significantly elevated O_2_^•−^ levels in their leaves, indicating the existence of cellular redox homeostasis imbalance and damage of important subcellular structures caused by MC-LR exposure, e.g., cell membranes of mesophyll cells [1].

### 2.3. Effects of MC-LR Exposure on GSH Levels in Rice Leaves

Figure 2 depicts the changes of GSH content in rice leaves following MC-LR treatment. The GSH content in rice leaves was significantly greater after 7 days of exposure to 50.0 μg·L^−1^ MC-LR compared with the control (*p* < 0.05). On the contrary, exposure to 50.0 μg·L^−1^ MC-LR for 15, 20, and 34 days resulted in significantly reduced GSH levels in rice leaves compared with the control (*p* < 0.05, *p* < 0.01). Furthermore, exposure to 0.10 μg·L^−1^ MC-LR for 34 days significantly increased the GSH content in rice leaves, whereas exposure to higher MC-LR concentrations of 1.0 and 10.0 μg·L^−1^ resulted in a significant reduction in GSH.

Numerous studies have demonstrated through in vitro cell-free systems or in vivo that, as the most abundant cellular thiol involved in the removal of ROS, GSH also plays an important role in the detoxification of MCs [1,35]. In the present study, there was a significant decrease in GSH content in the leaves of rice plants exposed to high concentrations of MC-LR after 15, 20, and 34 days of treatment (50.0 μg·L^−1^), indicating that GSH is involved in MC-LR detoxification, most likely by binding to MC-LR, increasing its solubility, and aiding in its removal from the organism. In addition, oxidative stress causes GSH to convert to the oxidized GSSG form, resulting in a decrease of GSH. As shown in Figure 2, after undergoing MC-LR stress for 34 days, the toxin at a low concentration (0.10 μg·L^−1^) can also stimulate the synthesis of GSH. Thus, changes in GSH content are the result of a comprehensive systemic response.

### 2.4. Effects of MC-LR Exposure on MDA Levels in Rice Leaves

As a marker for lipid peroxidation (LPO), the intracellular MDA content changes when the plant is exposed to environmental stress. As shown in Figure 3, at the lowest concentration of MC-LR (0.10 μg·L^−1^), there were no significant changes to the MDA content in rice leaves, regardless of length of exposure. After 15- and 34-days of exposure to 1.0, 10.0, and 50.0 μg·L^−1^ MC-LR, the MDA content in rice leaves decreased significantly compared with the control (*p* < 0.05, *p* < 0.01). However, the MDA concentration reduced significantly only in the 50.0 μg·L^−1^ group after 20 days of exposure (*p* < 0.05).

MDA is usually an important indicator of environmentally induced oxidative stress. When rice plants experience other stresses, including cadmium, mercury, or water changes, an increase in the level of MDA is observed [36,37]. Similar to this, the rise in LPO levels is a key factor in MC-LR-induced toxicity in aquatic plants. According to Jiang et al. [1], *Vallisneria natans* exposed to environmental concentrations of MC-LR showed a considerably higher MDA content. However, the current study showed that, after MC-LR exposure, significantly decreased MDA contents were observed in several groups. This observation may be related to the intimate relationship between the level of ROS and intracellular structural damages to a certain extent. This result ties in well with previous studies, wherein a reduction in MDA and ROS levels found in the aquatic macrophyte *Vallisneria natans* under relatively high-level MC-LR exposure was associated with the extent of damages to cellular organs like chloroplasts and mitochondria [1]. MC-LR has been shown to have the ability to disrupt the mitochondrial electron transport chain, thereby affecting the production of ROS. The damages to the cellular lipid membrane structure may result in a depletion of lipid substrated for LPO, and a depletion of intracellular GSH also occurs in cells, altering the intracellular redox status.

### 2.5. Effects of MC-LR Exposure on the Activity of Sucrose Synthase (SS) in Rice Leaves

As depicted in Figure 4, the activity of SS in rice leaves significantly decreased after exposure to 1.0 and 10.0 μg·L^−1^ MC-LR for 7 days compared with the control (*p* < 0.05). The 50.0 μg·L^−1^ MC-LR treatment group’s SS activity in rice leaves increased significantly on day 15 compared with the control group (*p* < 0.05). After 20 days exposure, the activity of SS reduced significantly in the 0.10 μg·L^−1^ MC-LR group in comparison with the control, whereas it increased significantly in the 1.0 μg·L^−1^ group (*p* < 0.05). After 34 days, the activity of SS was significantly lower in the 0.10, 10.0, and 50.0 μg·L^−1^ MC-LR groups in comparison with the control (*p* < 0.05).

MCs can specifically inhibit the activity of intracellular PP1 and PP2A, thereby activating protein kinase and cyclooxygenase to promote the phosphorylation of a variety of intracellular proteins. This disrupts the equilibrium between intracellular protein phosphorylation and dephosphorylation, inhibits intracellular signal transduction, and leads to a range of other physiological and biochemical responses that ultimately result in cell damage [4,20]. In the late stage of exposure (34 days), the activity of SS saw an overall decrease compared with the control, particularly in the groups exposed to 0.10, 10.0, and 50.0 μg·L^−1^ MC-LR. Lower SS activity may influence cell differentiation and cell wall synthesis and can reduce plants’ ability to accumulate carbon by decreasing the plants’ resistance. This is consistent with what was found in a previous study that the vegetative growth of rice seedlings exposed to MC-LR for 34 days was significantly reduced [10].

### 2.6. Effects of MC-LR on iNOS and TNOS in Rice Leaves

The activity of iNOS in the leaves of rice plants treated with MC-LR is depicted in Figure 5a. With exposure to 50.0 μg·L^−1^ MC-LR for 7 and 15 days, the iNOS activity significantly increased compared with the control (*p* < 0.05). After 34 days, the group treated with 0.1 μg·L^−1^ MC-LR showed significantly higher iNOS activity in comparison with the control group. In contrast, 10.0 and 50.0 μg·L^−1^ MC-LR exposure significantly lowered the iNOS activity compared with the control (*p* < 0.05, *p* < 0.01). There were no significant differences in total nitric oxide synthase (TNOS) activities when rice was exposed to varied concentrations of MC-LR for 20 days compared with the control. The TNOS activity in plants treated with MC-LR is shown in Figure 5b. Following the exposure to 50.0 μg·L^−1^ MC-LR for 7 and 20 days, the TNOS activity in rice leaves was significantly induced in comparison with the control (*p* < 0.05). After being exposed to 0.10 μg·L^−1^ MC-LR for 34 days, the TNOS activity was significantly greater than the control. In contrast, the TNOS activity was significantly lower after exposure to 10.0 and 50.0 μg·L^−1^ MC-LR (*p* < 0.05, *p* < 0.01). There were no remarkable differences in the TNOS activity when exposed to various concentrations of MC-LR for 15 days compared with the control. A significant positive correlation was found between iNOS and TNOS activity in rice leaves under MC-LR exposure (Figure 6).

Nearly all plant physiological processes involve nitric oxide (NO), a signaling free radical, either directly or indirectly. Although L-arginine (L-Arg)-dependent nitric oxide synthase (NOS), the enzymatic NO source in animal systems, has been well defined, it is still unclear how NO is produced enzymatically in higher plants [38]. NO interacts in different ways with ROS and may play a role as an antioxidant under certain stress conditions [25,26,27]. Moreover, NO can also modulate superoxide formation and inhibit LPO [27,39]. The reaction between NO and O_2_^•−^, can generate ONOO-, which can break down into the strongly oxidizing and strongly cytotoxic OH• and NO_2_ molecules that do damage to cells. Therefore, a balance between ROS and NO is very important. NO can also modulate root morphology and growth by regulating auxins when under stress from MC-LR [26,27]. Importantly, NO is mainly synthesized and released by NOS and nitrate reductase, among which iNOS plays a dominant transcriptional role and synthesizes most of the cellular NO [28]. Under stress, most of the NO production in plant tissues is dependent on the NOS pathway. Huang et al. [40] have shown that auxin indole-3-butyric acid (IBA) induces the biosynthesis of NO by NOS in adventitious roots. Ji et al. [28] discovered in a different investigation that MC-LR can increase the iNOS protein levels in rat pancreatic tumor cell lines. Similar outcomes are reported in the current investigation. Rice leaves significantly increased iNOS activity after being exposed to 50.0 μg·L^−1^ MC-LR for 7 and 15 days. Both iNOS and TNOS considerably increased in rice leaves after 34 days of exposure to a low concentration (0.10 μg·L^−1^) of MC-LR. After being exposed to 10 and 50 μg·L^−1^ MC-LR for 34 days, iNOS and TNOS concentrations were significantly lower, which echoes the findings of Chen et al. [13], who found that the NO content was significantly reduced in rice exposed to MC-LR for 2 days. NOS also plays a positive role in dephosphorylation. NO-mediated apoptosis can also be triggered by the rapid induction of iNOS, and additionally causes DNA damage that needs to be further studied [28].

### 2.7. Correlation between Physiological and Biochemical Indexes in Rice

This study shows that exposure to dissolved MC-LR in hydroponic systems results in the accumulation of toxins in rice leaves and roots, which can cause a range of physiological and biochemical changes in plant leaves. The correlation between the changes of these physiological and biochemical indicators in rice plants is illustrated in Figure 6. After exposure, there was a significant negative correlation between GSH content in rice leaves and soluble MC-LR concentration in hydroponic systems. Additionally, iNOS and TNOS activities exhibited a strong positive correlation with GSH content in leaves. The correlation between all the physiological and biochemical indexes in leaves and the accumulation of toxin in leaves and roots was not significant in the current study. The normal cellular redox state in plants is represented by the amount of GSH content and the NOS-related enzymes’ activities, and it is strongly correlated with the degree of oxidative stress in organisms under environmental stress. Similar correlations have been shown in other similar studies, e.g., MDA levels were found to be strong correlated with ROS levels in both aquatic and terrestrial plants [1,41,42]. When comparing our results with those of other studies, it must be pointed out that cellular GSH is a potential biomarker for algal toxin stress in the environment as it has an important role in organisms’ resistance to MC toxicity. However, due to the active resistance to MCs and the specific excretion mechanism, it may be difficult to compare environmental MCs’ pollution levels with the accumulation levels of MCs in plants.

## 3. Conclusions

The exposure to dissolved MC-LR in culture solution at 1.0–50.0 μg·L^−1^ resulted in the accumulation of toxins in rice leaves and roots, which can cause a range of physiological and biochemical responses. However, a significant accumulation trend of MC-LR in plants (BCF > 1) was only found in the lowest group (0.10 μg·L^−1^). At the beginning of exposure (10 days), the levels of O_2_^•−^ were significantly reduced. As the exposure time increased to 20 days, the levels of O_2_^•−^ became induced significantly. The initial reduction of O_2_^•−^ content in rice leaves may take place through the induction of the antioxidant system and of NOS activity. Intracellular NO production could be indirectly affected by MC-LR because the iNOS and TNOS activities were significantly affected by different concentrations of MC-LR exposure. The MDA content was significantly lower in plants exposed to relatively higher concentrations of MC-LR (1.0, 10.0, and 50.0 μg·L^−1^). This observation may be closely related to the level of free radical damage and changes in intracellular membrane structures in rice leaves. The inhibition of vegetative growth was also supported by the observation of the overall decreased activity of SS. The changes in GSH content in rice leaves could be recognized in a dualistic pattern, which showed positive correlation with the changes in iNOS and TNOS activities. Taken together, these changes of physiological and biochemical indexes imply that the toxicity consequences of prolonged exposure to low concentrations of MC-LR are more complicated and warrant more investigation in the future. In addition, there is a tremendous difference between hydroponics and soil cultivation in exposure. Therefore, methods for analyzing actual rice growth conditions more closely in order to ascertain more realistic physiological reactions in plants due to environmental stress are thus worthy of further investigation.

## 4. Materials and Methods

### 4.1. Chemicals and Reagents

Microcystin-LR, ≥95% in purity (HPLC) was purchased from Taiwan Algae Institute Ltd. (Taoyuan City, Taiwan). The Microcystin Plate Kit was purchased from the Institute of Hydrobiology, Chinese Academy of Sciences. Phenylmethylsulfonyl fluoride (PMSF), o-phthaldialdehyde, bovine serum albumin (BSA), polyvinylpolypyrrolidone (PVPP), and α-naphthylamine were purchased from Sigma-Aldrich (St. Louis, MO, USA). Methanol, acetic acid, and dimethyl sulfoxide (DMSF) were purchased from Merck (Darmstadt, Germany). Other reagents were analytical grade and purchased from Chinese companies.

### 4.2. Plant Material and Exposure Experiments

The Nipponbare rice variety (*Oryza sativa* L.) was used for the following experiments. One thousand robust and full Nipponbare rice seeds were selected and divided into four groups. The seeds were soaked at 28 °C for 24 h and germination was induced by covering the rice seeds with three layers of wet gauze and placing them into containers lined with aluminum foil, so that the seeds would be in a dark and humid environment. The container was placed in a 25 °C incubator to induce germination and the seeds were kept in a humid environment throughout the germination process. Rice seedlings were transferred to the International Rice Research Institute in conventional nutrient solution after germination (containing 40 mg·L^−1^ Na^+^, 40 mg·L^−1^ K^+^, 40 mg·L^−1^ Ca^2+^, 40 mg·L^−1^ Mg^2+^, 10 mg·L^−1^ P^5+^, 2 mg·L^−1^ Fe^3+^, 0.5 mg·L^−1^ Mn^2+^, 0.05 mg·L^−1^ Mo^6+^, 0.2 mg·L^−1^ B^3+^, 0.01 mg·L^−1^ Zn^2+^, and 0.01 mg·L^−1^ Cu^2+^) and cultured.

Plant seedlings were cultured in nutrient solution until 5 days after germination. Plants that were growing well were selected, cleaned with ultrapure water, and subjected to the MC-LR exposure test. The test groups were exposed to MC-LR concentrations of 0.10, 1.00, 10.0, and 50.0 μg·L^−1^, while a negative control group was exposed to nutrient solution without MC-LR. All seedlings were exposed for 7, 15, 20, or 34 days, and four parallel subgroups with 10 plants were defined in each group. During the exposure period, the nutrient solution containing MC-LR was replaced every other day. The initial solution was 100 mL of diluted nutrient solution, but, with plant growth, the dilution was gradually reduced and the nutrient solution was increased to 200 mL to ensure that the plants had sufficient nutrients for growth. At the end of the exposure period, the plant biomass was measured immediately. Fresh leaves and roots were taken, quick-frozen in liquid nitrogen, packaged, and kept in a −80 °C freezer.

### 4.3. Microcystin Analysis

MC-LR in plants was determined according to Jiang et al. [1]. Leaf tissues were ground to fine powder under liquid nitrogen, and then extracted with 5% acetic acid followed by 90% methanol. The extracts were pooled, centrifuged at 15,000 rpm for 20 min, and diluted with Milli-Q water before being applied to Oasis HLB cartridges (Waters, Ireland). The MC-LR in the cartridges was eluted with 90% methanol and evaporated to dryness. The dried exacts were reconstituted with Milli-Q water and analyzed by ELISA (Microcystin plate kit, Institute of Hydrobiology, Chinese Academy of Sciences). Extraction and centrifugation were conducted at 4–8 °C.

### 4.4. Determination of Superoxide Anion (O_2_^•−^) Content

On days 10, 20, and 34 after exposure to MC-LR, about 0.2 g fresh leaves were collected and accurately weighed, then quickly pulverized to powder under liquid nitrogen. A five-fold volume of 50 mmol·L^−1^ phosphate buffer (pH 7.8) containing 0.1 mmol·L^−1^ EDTA, 4% (*w*/*v*) PVPP, and 0.3% (*w*/*v*) Triton X-100 was added to make a homogenate, which was centrifuged at 10,500× *g* for 20 min. Immediately, 0.5 mL of the supernatant was drawn and added to 0.5 mL of phosphate buffer (50 mmol·L^−1^, pH 7.8) and 1 mL of 1.0 mmol·L^−1^ hydroxylamine hydrochloride. The mixture was shook to mix before being incubated at 25 °C for 1 h. One milliliter of 17.0 mmol·L^−1^ sulfanilamide (prepared with glacial acetic acid and water in a 3:1 ratio) and 1 mL of 7.0 mmol/L α-naphthylamine (prepared with glacial acetic acid and water in a 3:1 ratio) were added. After mixing, the sample was incubated at 25 °C for 20 min before its absorbance at 530 nm was measured. Using a solution of sodium nitrite, a standard curve was generated, as described above, and the O_2_^•−^ content was calculated based on the standard curve. 

### 4.5. Enzyme Extraction and Activity Assays

About 0.2 g fresh leaves were collected and quickly pulverized to a fine powder under liquid nitrogen. The extraction of crude enzyme from the powder was performed according to Jiang et al. [1]. The extracts was separated into aliquots and kept at 80 °C for additional investigation. All operations were carried out at 0–4 °C. The determination of sucrose synthase (SS, EC 2.4.1.13), inducible nitric oxide synthase (iNOS), and total nitric oxide synthase (TNOS, EC 1.14.13.39) activities was performed using a commercialized biochemical assay kit (Nanjing Jiancheng Bioengineering Institute, Nanjing, China), according to the manufacturer’s instructions. In all cases, the total protein (Pr) content was measured according to the Bradford method [43], with bovine serum albumin as a standard.

### 4.6. Glutathione and Malondialdehyde Determination

The amounts of GSH in plant leaves were measured according to Hissin and Hilf [44], with modifications according to Jiang et al. [1]. Frozen leaf tissues were extracted with 100 mmol·L^−1^ sodium phosphate-EDTA buffer (pH 8.0) and 6.5% trichloroacetic acid. The homogenates were centrifuged at 12,000 rpm for 20 min at 4 °C. GSH levels in the supernatant were measured fluorometrically after incubation with o-phthaldialdehyde in phosphate-EDTA buffer. The fluorescence intensity was recorded at 420 nm after excitation at 350 nm on a fluorescence spectrophotometer (Hitachi, Ichige, Hitachinaka, Japan).

The amounts of MDA in plant leaves were determined using a biochemical assay kit (Nanjing Jiancheng Bioengineering Institute, Nanjing, China), according to the manufacturer’s instructions.

### 4.7. Data Analysis

Data were checked for normality using the Shapiro–Wilk test, and, when necessary, they were converted to adhere to the normal distribution assumption. With the use of SPSS 15.0 software, a one-way ANOVA test and Duncan’s multiple range test were employed to examine the statistical significance. Differences of *p* < 0.05 were considered significant, while *p* < 0.01 was considered highly significant. Quadruplicate analyses of each sample were conducted. The Pearson Correlation Coefficient between the two variables was calculated using the Correlation Plot tool in Origin 2021, and the correlation heat map was drawn.

## Figures and Tables

**Figure 1 toxins-16-00082-f001:**
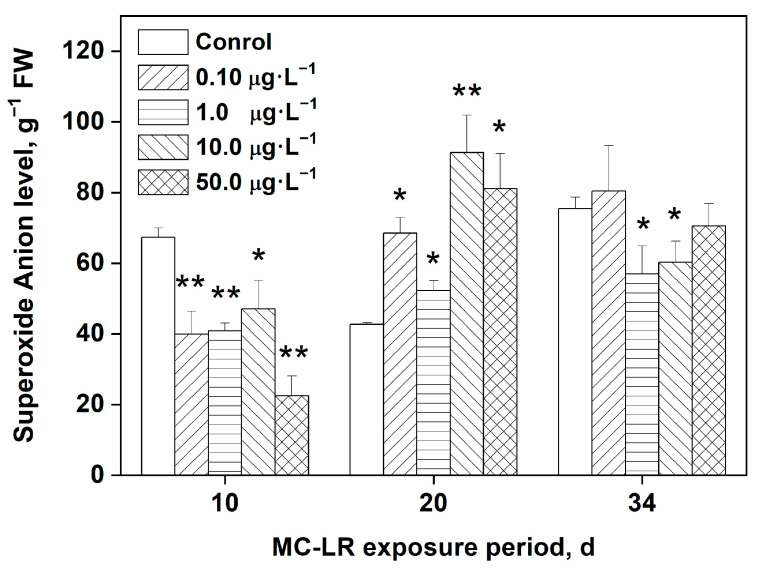
Levels of the superoxide anion, O_2_^•−^, in rice leaves after MC-LR exposure. Data are denoted as mean ± standard deviation (n = 4). Significant differences from the control group at each time point are indicated as * (*p* < 0.05) and ** (*p* < 0.01). The same for the following figures.

**Figure 2 toxins-16-00082-f002:**
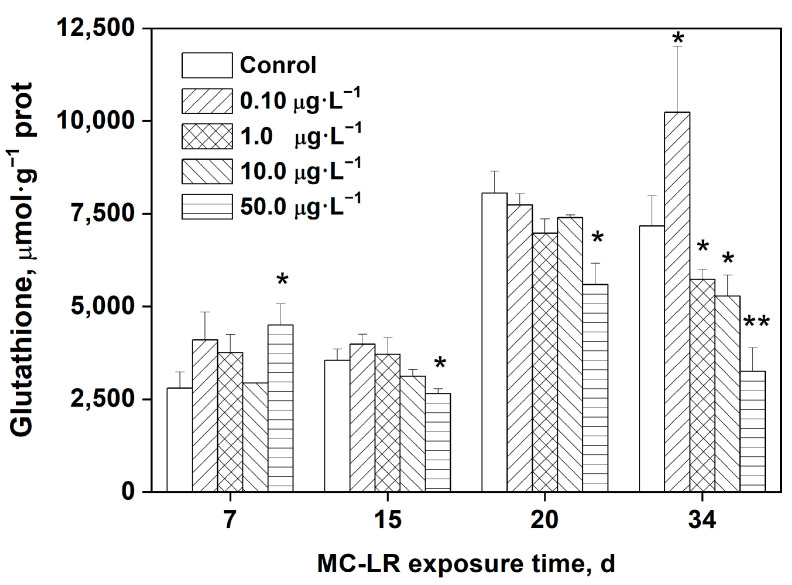
Levels of GSH in the leaves of rice seedling exposed to MC-LR. Significant differences from the control group at each time point are indicated as * (*p* < 0.05) and ** (*p* < 0.01).

**Figure 3 toxins-16-00082-f003:**
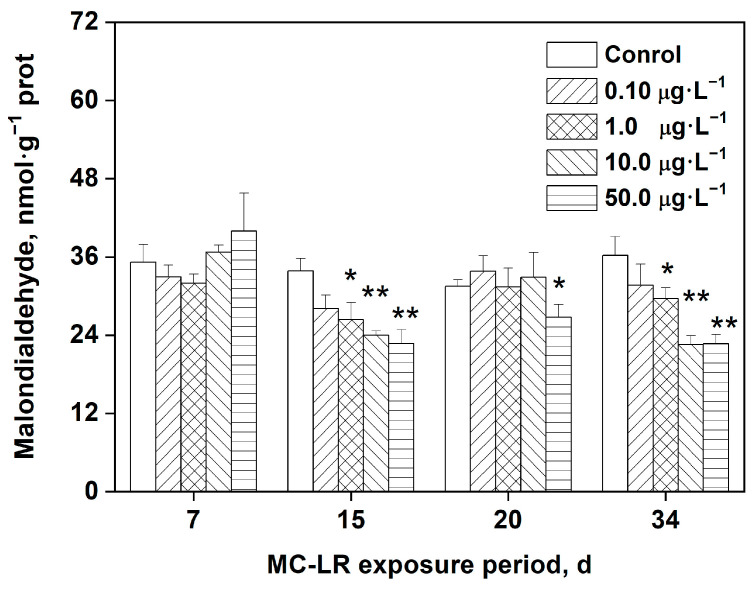
Levels of MDA in the leaves of rice exposed to MC-LR. Significant differences from the control group at each time point are indicated as * (*p* < 0.05) and ** (*p* < 0.01).

**Figure 4 toxins-16-00082-f004:**
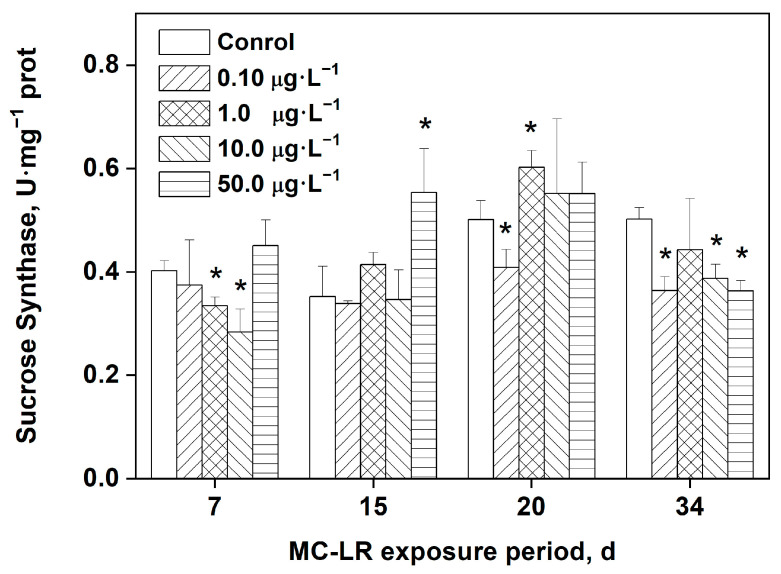
Activity of SS in the leaves of rice exposed to MC-LR. Significant differences from the control group at each time point are indicated as * (*p* < 0.05).

**Figure 5 toxins-16-00082-f005:**
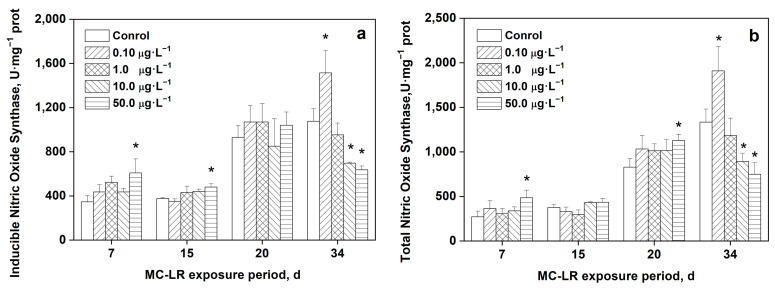
Activity of iNOS (**a**) and TNOS (**b**) in the leaves of rice seedlings exposed to MC-LR. Significant differences from the control group at each time point are indicated as * (*p* < 0.05).

**Figure 6 toxins-16-00082-f006:**
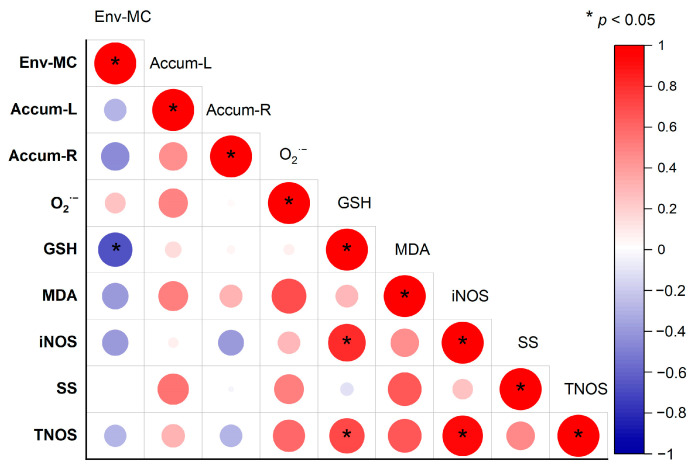
Correlation between physiological and biochemical indexes in rice after exposure to MC-LR. The size of the circle and the color depth in the Correlation Heat Map represent the size of the Pearson Correlation Coefficient; * indicates the correlation is significant (*p* ≤ 0.05). Env-MC: MC-LR concentrations in the nutrient solution; Accum-L: MC-LR contents in rice leaves; Accum-R: MC-LR contents in rice roots.

**Table 1 toxins-16-00082-t001:** Bioaccumulation of MC-LR in rice leaves and roots.

MC-LR Concentration, μg/L	Bioaccumulation of MC-LR in Rice Leaves and Roots									
	Leaves, μg/g FW					Roots, μg/g FW				
	Exposure Period *, Days				BCF **	Exposure Period, Days				BCF
	7	15	20	34		7	15	20	34	
0.1	n.a. ***	0.54 ± 0.09 bcde ****	0.59 ± 0.08 abcde	0.51 ± 0.09 cde	5.90 ± 0.80	0.68 ± 0.05 de	0.77 ± 0.04 bcd	0.75 ± 0.14 bcd	0.67 ± 0.05 de	7.70 ± 0.40
1.0	n.a.	0.67 ± 0.06 abcd	0.69 ± 0.04 abcd	0.44 ± 0.10 de	0.69 ± 0.042	0.68 ± 0.05 de	0.72 ± 0.05 cd	0.75 ± 0.03 cd	0.71 ± 0.19 cde	0.75 ± 0.03
10.0	n.a.	0.85 ± 0.17 ab	0.88 ± 0.11 a	0.78 ± 0.06 abc	0.088 ± 0.011	1.12 ± 0.28 bc	1.16 ± 0.16 ab	0.98 ± 0.10 bcd	0.83 ± 0.19 bcde	0.12 ± 0.02
50.0	n.a.	n.a.	0.64 ± 0.46 abcde	0.33 ± 0.07 e	0.013 ± 0.01	1.55 ± 0.52 a	0.58 ± 0.05 de	0.46 ± 0.39 e	0.74 ± 0.21 cde	0.03 ± 0.01

Data are denoted as mean ± standard deviation (n = 3). * We did not collect enough leaves for determination of MC-LR content; ** Here, BCF stands for the maximum ratio of MC-LR in rice leaves or roots to the aqueous MC-LR concentration; *** n.a. = not available; **** Different letters indicated that the data points are significantly different at *p* < 0.05.

## Data Availability

The data presented in this study are available upon request from the corresponding author.

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
