# Peer review of "Bioaccumulation of Microcystin-LR and Induced Physio-Biochemical Changes in Rice (*Oryza sativa* L.) at Vegetative Stage under Hydroponic Culture Conditions"

_toxins, 2024, doi:10.3390/toxins16020082_

Round 1

Reviewer 1 Report

Comments and Suggestions for Authors

Title

Bioaccumulation of microcystin-LR and induced physio-bio- chemical changes in rice (Oryza sativa L.) at vegetative stage under hydroponic culture conditions

Review comments

The MS deals with an interesting research topic regarding the studies on the effects of cyanotoxins on physio-biochemical processes changes in the leaves of rice seedlings growing under hydroponic culture conditions.  The main objective focused on the study on the mechanism of microcystin-LR (MC-LR) induced phy- totoxicity in rice (Oryza sativa L.) at environmental concentrations

After a careful review of the manuscript and before giving the final decision about the status of the publication, certain scientific questions were raising in order to improve and make clearer the understanding of the content.

Major comments

Line 18: what do you mean by ‘at later exposure time’? Be more specific.

Lines 21-22: What do you mean by this information  ‘--implying that NO may play a role in MC-LR toxicity in rice’? NO is not a toxic compound for plants; it has several roles in regulating plant growth and enhancing tolerance to biotic and abiotic stressors.

Can you be more specific about the MC-LR concentrations that induced the observed physio-biochemical changes? For some indices, it is not clear at all, such as NOS and SS.  I was a bit confused while reading the outputs in the abstract.

Lines 32-33: to edit (grammatical error).

Lines 59-60: Please re-phrase.

Lines 78-79:  You wrote “ Current research on the impact of environmental concentrations of MC-LR accumu-77 lation in terrestrial plants is not extensive, nor is the research on the MCs induced physi-78 ological and biochemical responses in plants. » . I’m not agree with you, There are lots of works about MC-induced physio-biochemical changes. I do not believe that the data about that is scarce if that is what you mean ???.

-The introduction section needs to be revised and edited to correct syntax and grammar errors.

Lines 92-93: Did you conduct any statistical analysis for the results of Table 1? You should provide a two-way ANOVA analysis for these results. As for the statement here, I see no increase in MC-LR content in plant leaves at 50 µg/ L compared to the other concentrations. It is almost the same for all exposure periods for both leaves and roots; can you explain that?

Lines 100-101: Please rephrase.

Lines 102-108: Can you explain why the exposed plants accumulate less MC-LR at higher concentrations, as revealed by the BCF? 

Table 1: Statistical analysis? What does the letter ‘d’ mean in the third line of the table? Days? You should provide the results of a two-way ANOVA in the table to see if there is any significant difference between exposure periods and environmental concentrations.

Line 145: ‘were still’. Please correct.

Lines 170-177: not clear at all. It’s very generalized and doesn’t explain clearly why O2- ions were lower by 10 d, higher by 20 d, and then lower by 34 d compared to the control group. I suppose after 10 d of exposure, the antioxidative system was triggered to neutralize the excessive amounts of ROS and that’s why their level decreased by 34 d.

Line 219: LPO? Lipid Peroxidation? 

Lines 223-224: please explain more.

Line 297: in hydroponic system.

Line 299: changes and not indexes.

Lines 300-305: Explain these negative and positive correlations.

Lines 310-313: Rephrase!

Line 325: -- were significantly reduced.

Lines 326-327: edit this phrase.

Line 314: in exposure to MCs. Not complete.

In figure 1, after the plants have been exposed to MC for 10 days, we notice that the superoxide anion concentration is higher in the control group than in the MC-treated groups, i.e. the control plants are more stressed. In your opinion, is it right normal? Also, after plants have been exposed to MC for 20 days, the concentration of superoxide anion is higher in the group treated with 10 µg/L MC than in the groups treated with 50 µg/L MC. How do you explain this? Knowing that in hydroponic

culture, 10 µg/L is a very high concentration. Same remarks for figure 2 and 3 ..., generally MDA reflects lipid peroxidation of membranes and since research MDA increases when exposed to MCs. The same applies to enzymatic activity. If not, do you have any explanation for your results?

Materials and methods

Line 272: What do you mean by ‘every other day’? replaced on a daily basis? In a two-day interval?

Line 371: how many plants by subgroup?

Line 376: ‘In’ not ‘under’.

Line 378:

·         Please provide extra details about the extraction procedure for MCs from plant tissue. Was there any clean-up step using reverse-phase octadecyl-bonded silica cartridges? If yes, provide details.

·         What kind of ELISA test was used here? A competitive indirect ELISA?

·         Is it a congener-independent test for microcystins/nodularins or a congener-dependent test for MC-LR?

·         Did you use a MC-free tissue to evaluate the matrix effect during MC quantification with this kit? If yes, provide details.

·         Did you use a positive control? Provide details and confirm if the test was valid according to the positive control.

Please elaborate on a paragraph answering the above-mentioned questions. The reference you gave doesn’t provide any details about the ELISA test used, nor does it answer the questions above.

Line 383: after under?!

Line 389: what amino acid?

4.6 should be placed before 4.5

4.7: you do indeed need to conduct a two-way ANOVA to see the difference between concentrations of MC-LR, and between the concentration and exposure times.

References:

Please check references one by one and edit : missing info, species to write in italic, etc.

MCs are potent hepatotoxins with over 90 different variants [1,6]   please update the information using more recent references.

Comments on the Quality of English Language

-The introduction section needs to be revised and edited to correct syntax and grammar errors.

Reviewer 2 Report

Comments and Suggestions for Authors

Dear Editors/Authors,

The authors are reporting a study, which investigates the effects of MC-LR on different physio-biochemical processes in leaves and roots of rice seedlings under 34 days of exposure. Furthermore, the accumulation of MCs in leaf and root tissues was investigated by applying ELISA assays.

The manuscript gives interesting and innovative information on some under-investigated physio-biochemical markers of exposure to MCs, which an issue of utmost importance such as the discovery of new effects of the most abundant cyanotoxin worldwide (MC-LR) in agricultural plants. Important is also important to investigate the accumulation of this toxins in the tissues of as much as possible agricultural vegetables since this phenomenon seems to be species-specific. The scientific literature on MC-LR effects on plant performance and accumulation in tissues is scarce, especially when the exposure occurs at ecologically relevant concentrations and different development stages.  Nevertheless, the tissues analyzed are not edible, which would be crucial to conclude the risk to public health. Also, the methods applied in this manuscript to determine MCs in tissues do not allow certain conclusions to be drawn because they then require confirmation. Additionally, the manuscript does not always keep a logical sequence between the several topics addressed. Although the analytical method (ELISA) for accumulation determination is not the most suitable for the MCs determination in plants, the manuscript has innovative and suitable publishable data and will certainly be of interest to the readers of TOXINS.

However, major revisions will be necessary, and other suggestions could improve the manuscript.

Major comments:

Graphical abstract:?

Highlights: ?

Abstract:

L7-8 – There are already some studies that indicate the effects of MC-LR on rice plants, thus some toxicity effects of this toxin area have already been suggested. It is also incongruent with what is described in L45-46 and L55-62. The sentence must be reformulated. Still in this sentence please consider replacing the word ‘induced’ with ‘that induces’.

The abstract must have the objective of the study.

Keywords: The acronym NOS must be written in full.

Introduction:

The Introduction is also too long with issues repeated throughout the text. Furthermore, there is a lack of logical sequence of the topics covered in the introduction. It should be rewritten.

L35-36 – Please consider changing ‘include species from the genera.

L54 –The citation to the authors must include Chen et al.,.

L69-70 – In my opinion, this sentence must be removed, since in the introduction any studies with high MCs concentration were cited.

L71-72 – Please correct ROS, first the designation then the acronym.

L74 – Please define iNOS

L82-85 – How did you analyze the characteristics of absorption? Why did you consider that 34 days of exposure is a long-term exposure?

In the introduction, there is no mention of Reduced sucrose synthase, but in my opinion, a brief explanation of this must be presented.

Results and Discussion:

L112-141 – The results require further discussion. These results are not founded in the light of the other studies, regarding the profile of MC-LR accumulation and the comparison with the MC-LR accumulation in other plants. Please improve.

L155-156 – the information is repeated. Here it should be presented the meaning of * and **. The same for the following figures.

L175-177 – Is this concentration comparable with the ones studied?

L219 – Please define LPO.

L219 – Please define TNOS.

L284 – Please define IBA.

L290 – Please correct being exposure.

L296-315 – Is this correlation between physiological and biochemical indexes in rice taking into account the concentration-dependent response?

Conclusion

L329-331 – This is very speculative. Please reformulate.

Materials and Methods:

L378-380 – The methods applied must be presented here. How the authors control the degradation of MC-LR over the time of the exposure experiment? It is well known that MC-LR at low concentrations can be degraded at a higher rate by abiotic/biotic factors. This can have a tremendous effect on the BCF.

References:

The species name must be written in italics.

Some critical papers on this subject are missing. E.g., https://doi.org/ 10.3390/plants10040639

Comments on the Quality of English Language

Dear Editors/Authors,

In some cases, also the English written must be improved.

Reviewer 3 Report

Comments and Suggestions for Authors

This study shows that exposure to dissolved MC-LR in culture solution results in the accumulation of toxins in rice leaves and roots, which can cause a range of physiological and biochemical indexes in plant leaves. The correlation between the changes of these physiological and biochemical indicators in rice plants is discussed deeply in the manuscript. Particularly, a significant accumulation trend of MC-LR in plants (BCF > 1) was only found in the lowest group (0.10 μg·L-1). The manuscript is well written and the experimental design well thought. The materials and methods part require some additional information. I recommend this paper for publication after minor revisions.

Specific comments: 

Abstract is too long (335 words). Maximum is 200

Line 37: “Microcystis producing toxic secondary metabolites”, I think that the verb here should be “produce”

Line 54: “.Chen found that MC-LR..”, maybe “Chen and coworkers found that MC-LR…” is more appropriate

Line 59: “A widely acceptable mechanism MCs specifically” maybe “MCs specifically inhibit…” is more appropriate and less wordy

Line 66: What GSH stands for?

Results section:

Paragraph 2.1: If I understood correctly, it was not possible to detect MC-LR in the 7-day group and the 15-day group exposed to 50.0 μg/L MC-LR. And it is mentioned that the probable cause it the insufficient amount of leaves collected. Do you always collect the same number of leaves? Because maybe it can be that the absorbed amount was not detectable, under the limit of detection of the system. Additionally, I would add the column corresponding to 7 days of experiment, it gives entirety to the table. 

Line 140-141: I really liked how the discussion of this paragraph is constructed. I suggest to explain a bit why the damage of high concentrations of MC-LR can result in a decrease in accumulation. 

Line 146: “As a result”, maybe “for this reason” is more appropriate here. 

Line 151: “with increases of 60.7%, 27.7%, 114%, and 90.1%, respectively”, respectively to what? Each treatment has to be specified. 

Line 198: “It is also possible that a higher concentration 197 of MC-LR inhibits the synthesis of GSH” which explains the observations obtained in which condition? 

Line 199-200: “Therefore, there are many possible reasons behind the changes in GSH content”. I think that this sentence can be removed or moved at the beginning of the discussion regarding the changes of GSH content. 

Line 224: since there is a reference here (Jiang et al.), and the relationship between the level of intracellular structural damage and changes in ROS levels is mentioned few times, it would be nice to have it furtherly discussed here. What make them think that there is a relationship between the level of intracellular structural damage and changes in ROS levels? Why this is applicable here? 

Line 247-249: rephrase.

Materials and methods:

Line 352: No need to say it. The sentence can be removed.

Microcystin analysis: provide a brief description of the method. 

Glutathione and malondialdehyde determination: provide a brief description of the method. 

Line 407: describe briefly the method. 

Line 408: briefly describe the Bradford method.

References: Check format, is not equal for all of them. 

Round 2

Reviewer 1 Report

Comments and Suggestions for Authors

We have carefully followed the various responses and corrections made to the manuscript. It seems to me that all the corrections could have improved the scientific content of the work and made it clearer and more understandable.

So, at this stage of the revision, it seems to me that the manuscript meets, quite well,  the requirements for publication in Toxins, therefore, and if the other reviewers agree, this manuscript could be accepted for publication in Toxins.